


# The altitude of green OI 557.7 nm and blue $N_2^+$ 427.8 nm aurora

Daniel K. Whiter[1], Noora Partamies[2,3], Björn Gustavsson[4], and Kirsti Kauristie[5]

[1]School of Physics & Astronomy, University of Southampton, UK
[2]University Centre in Svalbard (UNIS), Longyearbyen, Norway
[3]Birkeland Centre for Space Science, Norway
[4]Dept. of Physics and Technology, UiT The Arctic University of Norway, Tromsø, Norway
[5]Finnish Meteorological Institute, Helsinki, Finland

**Correspondence:** D. K. Whiter (d.whiter@soton.ac.uk)

**Abstract.** We have performed a large statistical study of the peak emission altitude of green $O(^1D_2-^1S_0)$ (557.7 nm) and blue $N_2^+$ 1N (427.8 nm) aurora using observations from a network of all-sky cameras stationed across northern Finland and Sweden recorded during 7 winter seasons from 2000 to 2007. Both emissions were found to typically peak at about 114 km. The distribution of blue peak altitudes is more skewed than that for the green, and the mean peak emission altitudes were

$114.84 \pm 0.06$ km and $116.55 \pm 0.07$ km for green and blue emissions, respectively. We compare simultaneous measurements of the two emissions in combination with auroral modelling to investigate the emission production mechanisms.

During low energy electron precipitation ($<\sim$4 keV), when the two emissions peak above about 110 km, it is more likely for the green emission to peak below the blue emission than vice versa, with the difference between the two heights increasing with their average. Modelling has shown that under these conditions the dominant source of $O(^1S)$, the upper state of the green

line, is energy transfer from excited $N_2(A^3\Sigma_u^+)$, with a rate that depends on the product of the $N_2$ and O number densities. Since both number densities decrease to higher altitude, the production of $O(^1S)$ by energy transfer from $N_2$ peaks at lower altitude than the $N_2$ ionisation rate, which depends on the $N_2$ number density only. Consequently, the green aurora peaks below the blue aurora.

When the two emissions peak below about 110 km they typically peak at very similar altitude. The dominant source of

$O(^1S)$ at low altitudes must not be energy transfer from $N_2$, since the rate of that process peaks above the $N_2$ ionisation rate and blue emission due to quenching of the long-lived excited $N_2$ at low altitudes. Dissociative recombination of $O_2^+$ seems most likely to a major source at these low altitudes, but our model is unable to reproduce observations fully, suggesting there may be additional sources of $O(^1S)$ unaccounted for.

## 1 Introduction

The emission height of the aurora is often used in the derivation of other auroral or ionospheric parameters, such as horizontal spatial scales, velocities, or electron precipitation energies (e.g. Kaila, 1989; Knudsen et al., 2001; Kalmoni et al., 2017), or as a height of other measurements such as neutral wind and temperature (e.g. Griffin et al., 2006; Kosch et al., 2011; Billett et al., 2020). Usually an assumption must be made about the height of the aurora. These assumptions are typically based on obser-





vations conducted using instruments and techniques which are primitive by modern standards (Störmer, 1916; Harang, 1951;

Störmer, 1955). Often, earlier work measured the "lower border" of an auroral arc, which is dependent on the instrument sensitivity (as pointed out by Boyd et al. (1971) and Whiter et al. (2013)), and therefore results using such methods should be treated with caution. The present study was motivated by the need to improve estimates of the auroral peak emission height, and utilises tens of thousands of camera images for which a single peak emission height has been derived for each pair of coincident auroral images, enabling a large statistical study. In this work we analyse two of the brightest (and most observed)

auroral emissions; the OI 557.7 nm "green line" and the $N_2^+$ 427.8 nm "blue line".

The auroral green line is an atomic oxygen emission, from the $O(^1D_2-^1S_0)$ transition, while the blue line is a feature of the $N_2^+$ First Negative (1N) band system ($B^2\Sigma_u^+-X^2\Sigma_g^+$). However, despite coming from species with different number density altitude profiles, it has been known for some decades that the green and blue aurora have more similar height profiles than their parent species would suggest (Henriksen and Egeland, 1988). This similarity is thought to be because the usual primary route

by which $O(^1S)$ is produced in aurora is energy transfer from electronically excited $N_2$ in the $A^3\Sigma_u^+$ state (Henriksen, 1973; Sharp and Torr, 1979; Burns and Reid, 1984; Gerdjikova and Shepherd, 1987; Gattinger et al., 1996) (we hereafter abbreviate to $N_2(A)$). However, there are several other pathways producing $O(^1S)$ in aurora, including dissociative recombination of $O_2^+$, electron impact excitation of atomic O, and electron impact dissociation of $O_2$, and the balance of these (and other) pathways under different electron precipitation conditions is not yet well understood. In a companion study to this one, Partamies et al.

(2022) investigate variation in the peak emission height of green OI 557.7 nm aurora with magnetic local time (MLT) and solar wind driving, over both Lapland (nightside auroral oval) and Svalbard (dayside cusp, nightside poleward edge of oval). Here we present average peak heights for both emissions over Lapland, before examining the difference in height between coincident green and blue auroral emission in combination with modelling to investigate the emission production mechanisms. We examine statistics compiled from individual time instants treated as independent measurements; the temporal evolution of

auroral heights, for example over the course of specific events, will be considered in a separate study.

Section 2 describes the imaging instrumentation used in this work and briefly summarises the method of determining auroral peak emission heights developed by Whiter et al. (2013). Observational results are presented in Section 3. Section 4 explains how the green and blue line emissions are modelled, and a detailed comparison of model results with observations is presented in Section 5.

## 50  2  Instrumentation and image analysis

This work uses images from five ground-based all-sky cameras stationed across northern Finland and Sweden, which form part of the Magnetometers–Ionospheric Radar–All-sky Cameras Large Experiment (MIRACLE) instrument network (Syrjäsuo et al., 1998). The geographic locations of the stations used in this work are listed in Table 1. The fields of view of all of the cameras overlap, allowing determination of the auroral peak emission height using any pair of stations. The cameras operate

automatically throughout the winter, producing many tens of thousands of images per station per year. From 1996 to 2007 MIRACLE used primarily charge-coupled device (CCD) detectors equiped with an image intensifier (ICCD cameras) and



**Table 1.** Locations (geographic latitude and longitude) of the camera stations used in this work.

| Station | Geo. Lat. | Geo. Lon. |
|---|---|---|
| Abisko | 68.36°N | 18.82°E |
| Kevo | 69.76°N | 27.01°E |
| Kilpisjärvi | 69.02°N | 20.79°E |
| Muonio | 68.02°N | 23.53°E |
| Sodankylä | 67.42°N | 26.39°E |

narrow-passband interference filters to select specific auroral emissions, mounted within a filter wheel. This work has used images of OI 557.7 nm emission (green), which is the most dominant auroral emission, and $N_2^+$ 1N 427.8 nm emission (blue), recorded with the ICCD cameras. Green images are available for the full time period of ICCD observations, whereas blue

images are only available from September 2000 onwards. Only images recorded when the solar zenith angle at the camera station was larger than 101° and the IL index (a local analogue of the AL index computed from MIRACLE magnetometer data, Kauristie et al. (1996)) was less than −50 nT are used in this study, to exclude conditions of sunrise and sunset and to ensure that there was auroral activity.

     All MIRACLE ICCD camera images recorded between 2000 and 2007 (the period when at least 2 mainland stations were
available and both green and blue emissions were measured) have been analysed using two fully automated independent methods for computing the auroral peak emission height. These methods have been described in detail by Whiter et al. (2013). Both methods analyse a pair of coincident images from different stations with overlapping fields of view. One of the methods (the "field line method") involves projecting all-sky images onto many geomagnetic field lines, while the other method (the "horizontal plane method") involves projecting the images onto planes parallel to the Earth's surface at different heights. Since
aurora is typically taller than it is wide, in general the field line method gives a more accurate measure of the peak emission height, and all height data shown in this work were obtained using the field line method with a precision of 0.5 km. However, to ensure measurements are reliable only image pairs for which the two methods agree to within 20 km are included in the statistical analysis. Of these, there are 57,907 simultaneous measurements of the height of both green and blue aurora from the same station pair (since the cameras use filter wheels to select each individual emission, there is approximately 2 s between
the beginning of the green exposure and the beginning of the following blue exposure, which is comparable to the exposure time and short enough to consider the images to be simultaneous). The uncertainty of each individual height measurement is difficult to determine and varies with type of aurora and geometrical properties of the aurora, but is estimated to be ±1–10 km. Importantly, Whiter et al. (2013) showed that the field line method is not biased towards high or low altitudes, so the uncertainty is not systematic. A degradation of the instruments' image intensifiers would lead to a decrease in the minimum observable
radiance, but the methods used here to find the peak emission height do not depend on absolute intensity, and therefore it is possible to combine images from different instruments without significant bias.

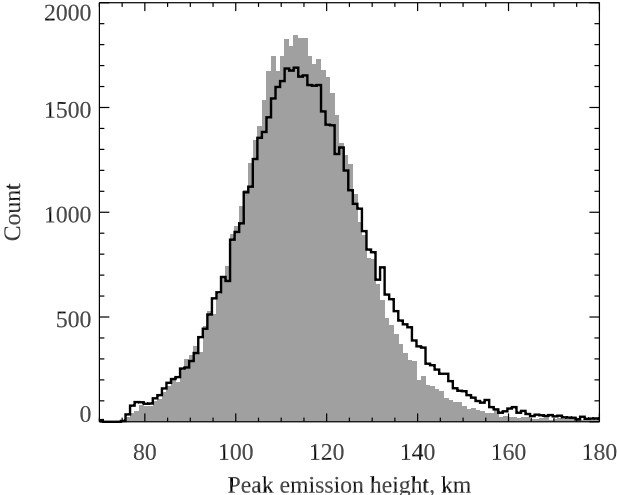

**Figure 1.** Histograms of observed peak emission heights for the green 557.7 nm emission (grey shading) and blue 427.8 nm emission (black outline).

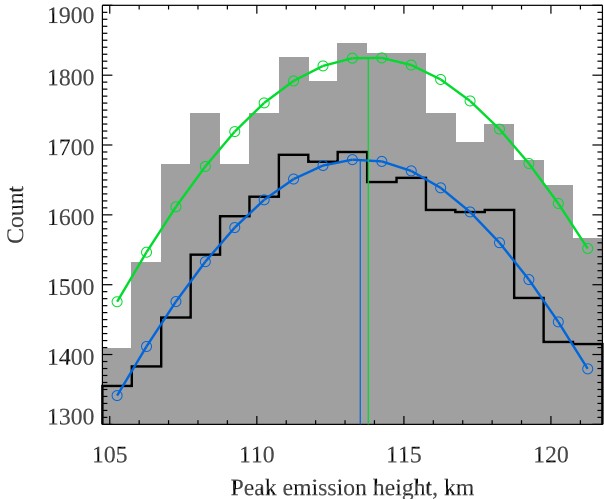

**Figure 2.** The peaks of the histograms as shown in Figure 1, together with Gaussian curves fitted to the histograms for the 557.7 nm emission (green) and 427.8 nm emission (blue). The vertical green and blue lines mark the peaks of the Gaussians.

## 3 Observational results

Figure 1 shows histograms of measured peak emission heights of green 557.7 nm aurora (shaded grey) and blue 427.8 nm aurora (black line), with a bin width of 1 km. Only measurements at times when height estimates of both emissions are available are included in the histograms, in order to allow a fair comparison. Both distributions peak close to 114 km, and have very



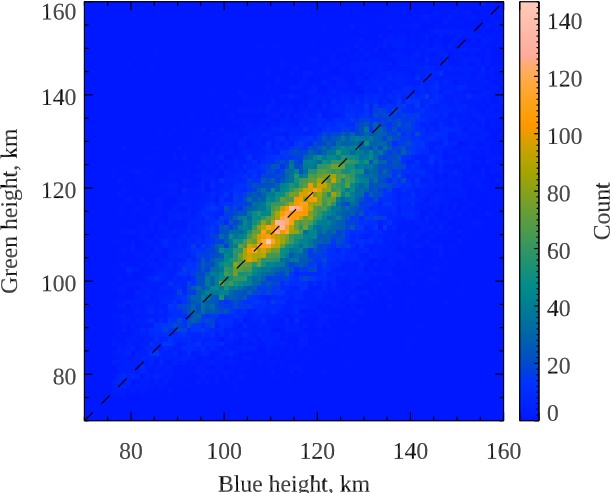

**Figure 3.** 2-dimensional histogram of simultaneous green (557.7 nm) and blue (427.8 nm) peak emission heights. The shading represents the number of observations in each histogram bin, according to the colour bar scale shown on the right of the figure. The dashed black line indicates where the two peak emission heights are equal.

similar tails towards lower peak emission heights. However, the distribution of green heights has a steeper tail towards greater peak emission heights than the distribution of blue heights. These histograms show that the blue line is almost twice as likely (factor 1.8) to peak at 140 km as the green line, although it is rare for either of these emissions to peak at such high altitudes. The median peak emission heights for green and blue aurora are 114.0 km and 115.0 km respectively (to nearest 0.5 km), and the corresponding mean heights are $114.84 \pm 0.06$ km and $116.55 \pm 0.07$ km, suggesting that the blue line emission peaks slightly above the green line emission. In order to find the maxima of the distributions, a Gaussian function was fitted to each of the histograms plotted in Figure 1 within the range 105.0–121.5 km (centred roughly on the peak of the distributions). Figure 2 shows the two histograms within this range, together with the fitted Gaussians in green (for 557.7 nm emission) and blue (for 427.8 nm emission). The coloured vertical lines within this figure show the maxima of the Gaussian functions, at 113.8 km for green aurora and 113.5 km for blue aurora. These results show that green and blue aurora peak at very similar altitudes, on average, but that it is more likely for blue to peak at high altitudes (above about 130 km) than green, contrary to a common misconception that blue peaks below green.

In order to compare the peak emission heights of coincident green and blue aurora, a two-dimensional histogram was created, shown in Figure 3. The bin width is 1 km for both green aurora (ordinate) and blue aurora (abscissa), and each 1 km × 1 km square is shaded according to the colour scale on the right of the figure to indicate the number of simultaneous height measurements of green and blue aurora within the ranges of that bin. The dashed diagonal line indicates where the two emissions peak at the same height ($y = x$ line). The bulk of the distribution between 105 km and 120 km closely follows this line, indicating that in general the two emissions peak at approximately the same altitude, as expected from the results

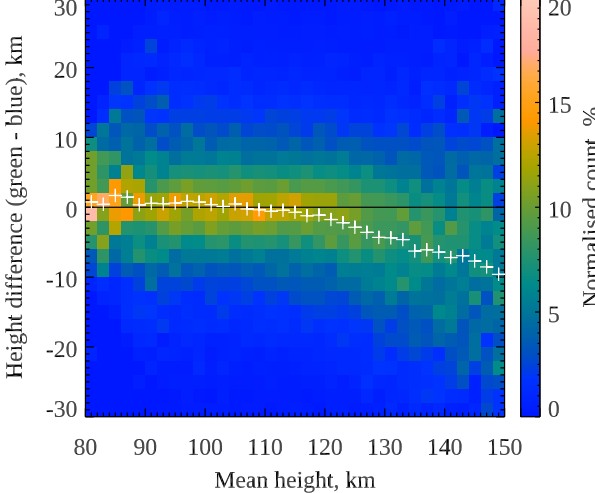

**Figure 4.** 2-dimensional histogram of auroral peak emission height observations binned according to the difference between green and blue heights (ordinate, positive values indicating green height above blue height) and mean (i.e. mid-point) of the simultaneous green and blue heights (abscissa). The shading represents the number of observations in each bin, normalised to the total number of observations in each column, according to the colour bar scale. The mean height difference for each column is shown as a white plus symbol.

presented above. At greater heights the majority of the distribution is below the dashed line, indicating that when the blue
emission peaks at relatively high altitude, the green emission peaks below it.

This result can be seen more clearly by plotting the difference in height between the green and blue emissions against the mean height (i.e. midpoint between the green and blue heights). Figure 4 shows a two-dimensional histogram with the height difference (positive values indicating that the green emission peaks above the blue emission) on the ordinate axis and the mean height on the abscissa axis. The histogram bins have a width of 2 km in each direction, and the colour scale indicates
the number of image pairs in each bin, normalised by the total number of image pairs in each column of the histogram. The white plus symbols show the mean height difference for each column of the histogram. As expected from the results shown in Figure 3, the height difference is close to zero for mean heights up to about 120 km. As the mean height increases beyond 120 km the average height difference becomes more negative, indicating that the blue emission peaks further above the green emission. The mean height was above 120 km in 34% of our observations.

## 3.1 Resonance scattering of sunlight by $N_2^+$

In sunlit conditions, $N_2^+$ 1N emission is produced through resonance absorption of solar photons by $N_2^+$ molecules in the ground state (Bates, 1949; Broadfoot, 1967). The resonance scattering contribution to the total blue line intensity can be significant, and typically occurs at higher altitude than the auroral emission produced by electron impact excitation (Jokiaho et al., 2009). To investigate whether the observed high peak altitude of the blue line could be caused by resonance scattering of sunlight,
histograms were created for the solar zenith angle and MLT at the time of the observations. Data were selected where the blue





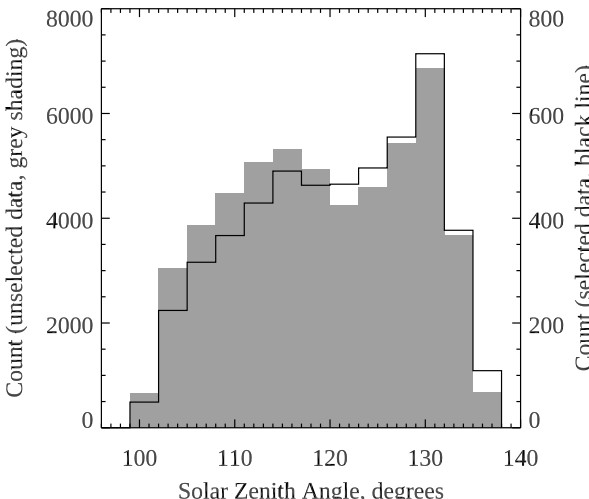

**Figure 5.** Histogram of the solar zenith angle at the time of the peak emission height measurements. The black line shows selected data (blue height above 130 km and at least 10 km above the green height) using the right ordinate axis, and the grey shading shows all remaining unselected data using the left ordinate axis.

line peaked above 130 km and at least 10 km above the coincident green line peak emission height, consisting of 8.7% of the total height measurements. Histograms were made separately for this selected data and all other unselected data, shown in Figure 5 (solar zenith angle) and Figure 6 (MLT). The solar zenith angle and MLT were calculated for the central latitude and longitude of the aurora, defined to be on the field lines used in the calculation of the peak emission height, rather than for the observation stations, to improve accuracy.

These histograms show that the high blue peak emission heights do not preferentially occur for low solar zenith angles or close to dawn or dusk, and therefore are not likely to be caused by resonance scattering of sunlight (or by daylight contamination on the horizon). The selected data are more centred around midnight MLT than the unselected data, but the two histograms shown in Figure 6 are not substantially different. See the companion paper by Partamies et al. (2022) for further discussion of the variation in height with MLT.

## 4  Auroral modelling

We use the Southampton ionospheric model as a tool to understand the observational results presented in Section 3. This model is a coupled electron transport (Lummerzheim and Lilensten, 1994) and time-dependent ion chemistry model (Lanchester et al., 2001) which calculates production and loss rates for various ion and neutral species as well as brightnesses of specific auroral emissions (Lanchester and Gustavsson, 2012; Whiter et al., 2012). The inputs to the model are an arbitrary electron precipitation spectrum and density profiles for the major neutrals (O, $N_2$ and $O_2$) taken from the NRLMSIS 2.0 model (Emmert et al.,

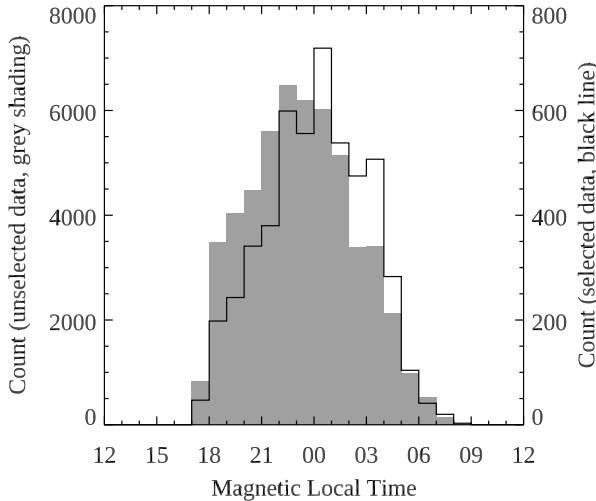

**Figure 6.** Histogram of the magnetic local time of the peak emission height measurements. The black line shows selected data (blue height above 130 km and at least 10 km above the green height) using the right ordinate axis, and the grey shading shows all remaining unselected data using the left ordinate axis.

2021), hereafter referred to as MSIS. As input to MSIS we use median conditions during the observations presented in Section 3, of $Ap = 17$, $F10.7 = 129$, 81-day average $F10.7 = 129$ and a time of 21:30 UT at Kilpisjärvi (69.02°N, 20.79°E).

The brightness of the blue line is calculated directly from the production rate of $N_2^+$ through electron impact ionisation
of neutral $N_2$. Since the emission is prompt, its brightness can be calculated directly from the ionisation-excitation rate. The lifetime of the $N_2^+$ 1N(0,1) band is 270 ns, as obtained from the Einstein coefficient given by Gilmore et al. (1992) (note this is long compared to the lifetime of the overall $N_2^+$ 1N band system but is appropriate for the 427.8 nm emission), which is too fast for quenching to have a significant effect on the peak emission height. Using quenching rate coefficients for the $N_2^+$ $B^2\Sigma_u^+$ state from Valk et al. (2010) together with $N_2$ and $O_2$ densities from the model and the quenching equation given by
Vallance Jones and Gattinger (1978), we estimate less than 1% of the blue emission is quenched at 90 km altitude.

The 557.7 nm green line is long-lived and therefore quenching is important, and so its intensity is calculated from the number density of the $O(^1S)$ upper state with an Einstein coefficient of $1.26\,\mathrm{s}^{-1}$ from Wiese et al. (1996). The reactions that produce $O(^1S)$ included in the model are listed in Table 2, together with references for their cross sections, rate coefficients and branching ratios, where relevant. Reactions R1 and R2 represent auroral electron impact on O and $O_2$, respectively. Reaction
R3 is dissociative recombination of $O_2^+$, and R4 is an $O_2^+$ loss reaction with atomic N. Reactions R5 and R6 represent energy transfer to atomic O from $N_2$ and $O_2$, respectively. The model includes losses of $O(^1S)$ through quenching by O and $O_2$ and losses through emission in 557.7 nm (lower state $O(^1D)$) and 297.2 nm (lower state $O(^3P)$). Further discussion of the rates relevant for the green line emission is included in the following section.





**Table 2.** Reactions producing O($^1$S) included in the Southampton ionospheric model.

|    | Reaction | References |
|----|----------|-----------|
| R1 | $O + e^- \rightarrow O(^1S) + e^-$ | Green and Stolarski (1972) |
| R2 | $O_2 + e^- \rightarrow O(^1S) + O + e^-$ | LeClair and McConkey (1993) |
| R3 | $O_2^+ + e^- \rightarrow O(^1S) + O$ | Sheehan and St.-Maurice (2004) |
|    |          | Kella et al. (1997) |
| R4 | $O_2^+ + N \rightarrow O(^1S) + NO^+$ | Gattinger et al. (1985) |
| R5 | $O + N_2(A) \rightarrow O(^1S) + N_2$ | Piper (1982) |
| R6 | $O + O_2^* \rightarrow O(^1S) + O_2$ | Gattinger et al. (1996) |

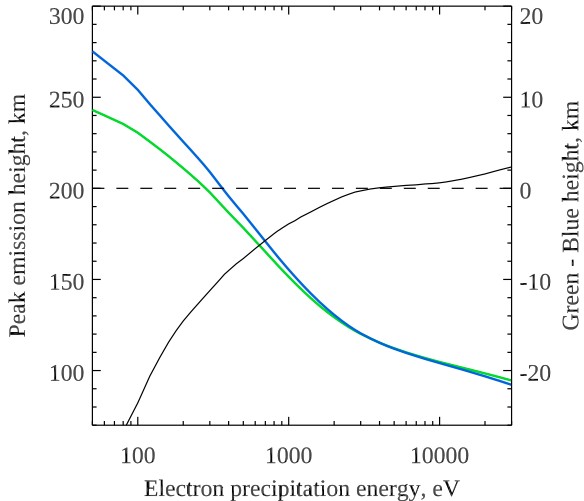

**Figure 7.** Peak emission heights of 557.7 nm aurora (green) and 427.8 nm aurora (blue) as a function of characteristic electron precipitation energy for Gaussian electron spectra, obtained using the Southampton ionospheric model. The difference in these heights is shown in black, using the right ordinate axis.

## 5 Model results and comparison with observations

Figures 7 and 8 show the modelled peak emission heights of the 557.7 nm (green) and 427.8 nm (blue) lines as a function of characteristic electron precipitation energy with a Gaussian (Figure 7) or Maxwellian (Figure 8) electron spectrum. The difference in these heights (blue height subtracted from green height) is also shown in black, according to the right ordinate axis in each figure. To determine these relationships the model was run independently at different characteristic energies with an electron energy flux of $20\,\mathrm{mWm^{-2}}$, and at each energy was run for a period of $12\,\mathrm{s}$ in order to reach a reasonably steady

state. Qualitatively, the model results resemble the observations presented in Section 3, in that the green and blue emissions



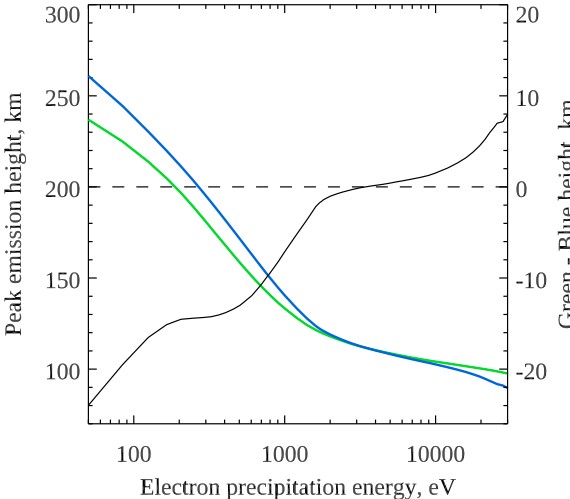

**Figure 8.** Peak emission heights of 557.7 nm aurora (green) and 427.8 nm aurora (blue) as a function of characteristic electron precipitation energy for Maxwellian electron spectra, obtained using the Southampton ionospheric model. The difference in these heights is shown in black, using the right ordinate axis.

peak at about the same height across much of the electron energy range tested, and at low energies, when the emissions peak at higher altitudes, the blue emission peaks above the green emission (more negative values on the right ordinate axis).

The model also does a good job of quantitatively reproducing the observations. For a quantitative comparison we return to the examination of how the height difference between the two emissions varies as a function of mean peak emission height, introduced earlier in Figure 4 and continued here in Figure 9, to avoid the need to estimate the electron precipitation energy during the observations. Since the peak emission heights are measured in the camera data to the nearest 0.5 km, the mean of the coincident green and blue peak heights always has a value divisible by 0.25 km. The average observed height differences calculated for possible values of the mean height (i.e. every 0.25 km) are shown as grey plus symbols in Figure 9. The average height difference calculated for 2 km wide bins of mean height, as shown as white plus symbols in Figure 4, are reproduced with black plus symbols in Figure 9. These black points show less scatter than the grey points since more observations are included in the calculation of each one. The solid blue and red curves represent the model results described above for Gaussian and Maxwellian spectra, respectively, and are essentially the same as the black curves shown in Figures 7 and 8 with the abscissae transformed from electron precipitation energy to mean height. The dashed blue (Gaussian) and red (Maxwellian) curves show the results of running the model for only 3 s instead of 12 s. We consider the results separately for low and high energy precipitation below.

## 5.1 Low energy electron precipitation

To understand the observations and model results we examined the altitude profiles of the relevant reaction rates and number densities. Figure 10 shows the modelled volume emission rate altitude profiles of the 557.7 nm and 427.8 nm emissions as





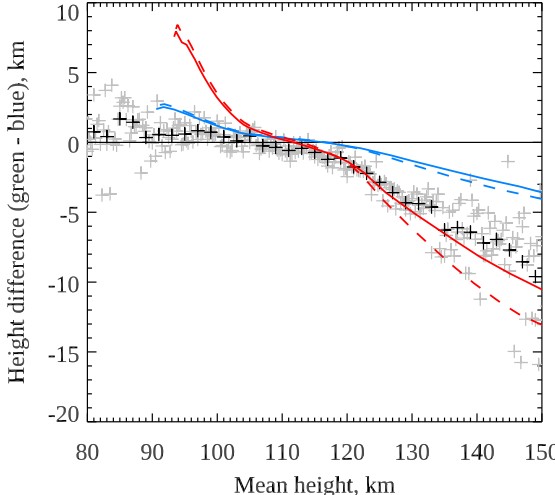

**Figure 9.** Observed and modelled difference in height between the green 557.7 nm and blue 427.8 nm emissions as a function of their mean height. Observations are shown as plus symbols. Grey symbols show the average height difference over all observations with the same mean height (to the nearest 0.25 km). Black symbols show the same, but where observations have been binned into 2 km wide bins of mean height (reproduced from Figure 4). The blue and red curves show the model results for Gaussian and Maxwellian shaped electron precipitation spectra, respectively, where the model is run for 12 s (steady state, solid curves) or for 3 s (dashed curves).

solid green and blue lines, respectively, for a Gaussian electron precipitation spectrum at 900 eV. The production rate of $O(^1S)$

by the two dominant sources during low energy precipitation, energy transfer from $N_2(A)$ (Reaction R5) and electron impact excitation (Reaction R1), are shown as dotted and dashed green lines, respectively, and the production rate of $N_2(A)$ is shown as a dotted black line. The peak emission heights of the two emissions are marked with horizontal lines. This figure highlights that at low electron precipitation energies the ionisation of $N_2$ peaks at a higher altitude than the production of $O(^1S)$ by energy transfer from $N_2(A)$, which could seem counter-intuitive given that the ionisation energy of $N_2$ is greater than the threshold

energy for excitation to $N_2(A)$. The reason is that the $N_2(A) + O$ energy transfer rate depends on the product of the atomic O number density and $N_2(A)$ number density. Since the O number density decreases to higher altitudes, this product peaks at lower altitude than the $N_2(A)$ density and $N_2^+$ production rate. Ultimately this is why the blue 427.8 nm emission peaks above the green 557.7 nm emission.

The model results using a Maxwellian electron spectrum best match the average observations above 110 km (dashed

red curve in Figure 9). However, the observations show broad variation in the height difference for high mean heights (see Figure 4), which is reproduced by varying the model run time and electron spectrum shape. Running the model for a shorter time period means that steady state is not achieved, in particular regarding the $N_2(A)$ and $O(^1S)$ number densities, and the spread of observed height differences may reflect the spatial and temporal dynamics of the auroral precipitation.



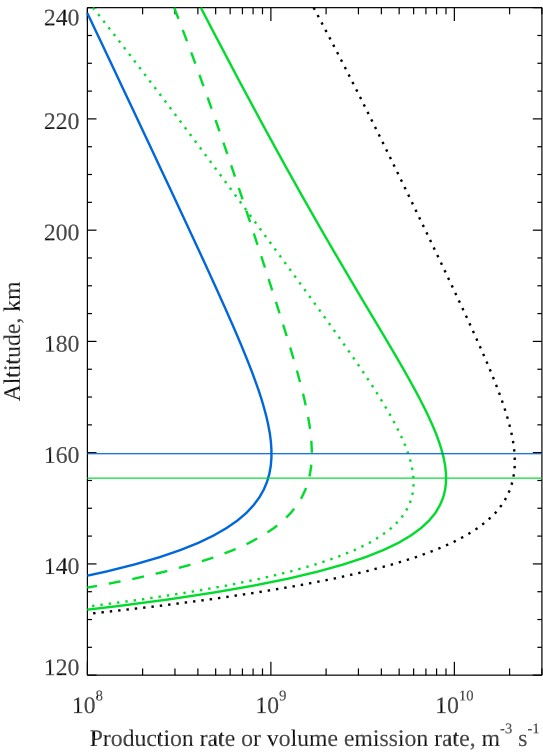

**Figure 10.** Modelled volume emission rate height profiles of the 427.8 nm blue line (solid blue) and 557.7 nm green line (solid green) using a 900 eV Gaussian electron precipitation spectrum. The horizontal lines mark the peaks of the volume emission rate profiles. Also shown are the production rates of $O(^1S)$ by energy transfer from $N_2(A)$ (dotted green) and electron impact on O (dashed green), and the production rate of $N_2(A)$ (dotted black).

## 5.2 High energy electron precipitation

For both Gaussian and Maxwellian electron spectra the model produces a green peak emission height above the blue height (i.e. positive height difference in Figure 9) for high energy precipitation where the mean peak emission height is below about 110 km, with the height difference increasing away from the observations with increasing energy of precipitation. The median $F10.7$ index at times where both emissions peak below 105 km is within 0.1 standard deviations of the median $F10.7$ index for other times, indicating that the low altitude emissions are primarily a result of energetic precipitation, rather than a contracted

atmosphere. At these high energies the Gaussian spectra produce model results that are more consistent with observations than those for the Maxwellian spectra, which may be an indication that more energetic precipitation (above about 4 keV) is typically more Gaussian in spectral shape, but neither spectral shape reproduces the observations well at the lowest altitudes. The mean height peaked below 110 km in 35% of our observations.

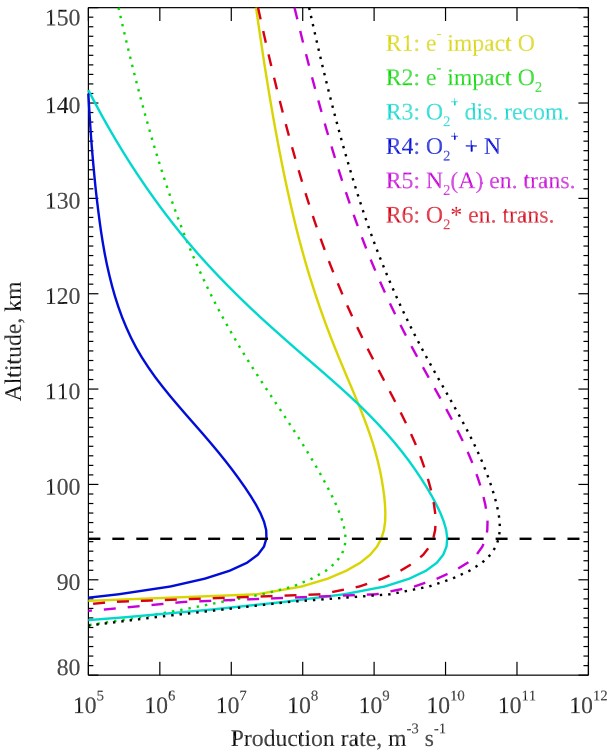

**Figure 11.** Modelled production rate profiles of O($^1$S) after 12 s of Gaussian electron precipitation at 25 keV. The reactions producing O($^1$S) are listed in Table 2. The rates of R1, R2, R3, R4, R5 and R6 are plotted in solid yellow, dotted green, solid cyan, solid blue, dashed magenta and dashed red, respectively. The total production rate of O($^1$S) is shown in dotted black. The horizontal dashed line marks the peak height of the blue 427.8 nm emission.

Figure 11 shows altitude profiles of the six O($^1$S) production mechanisms included in the model, for a Gaussian electron spectrum at 25 keV with a flux of 20 mW m$^{-2}$ running for 12 s. Auroral electron impact on O (Reaction R1) and O$_2$ (R2) are shown as solid yellow and dotted green lines, respectively. Dissociative recombination of O$_2^+$ (R3) is shown as a solid cyan line and Reaction R4 between O$_2^+$ and N is shown as a solid blue line. Energy transfer from N$_2$(A) (R5) and O$_2^*$ (R6) are shown as dashed magenta and red lines, respectively. The dotted black line shows the total production rate of O($^1$S) and the horizontal dashed line shows the peak height of the blue 427.8 nm emission. Of the six mechanisms, dissociative recombination of O$_2^+$, electron impact on O$_2$, and O$_2^+$ + N all peak at the same height as the 427.8 nm emission. Figure 12 shows the O$_2^+$ dissociative recombination rate peak height plotted against the 427.8 nm peak emission height for Gaussian electron precipitation. The dashed line represents equal rates ($y = x$). The two rates peak at very similar altitudes through the E-region ionosphere, especially below 105 km.

O'Neil et al. (1979) used rocket-based optical observations of artificial aurora with modelling to conclude that dissociative recombination of O$_2^+$ (R3) is the dominant production mechanism of O($^1$S) below 96 km, with energy transfer from N$_2$(A)

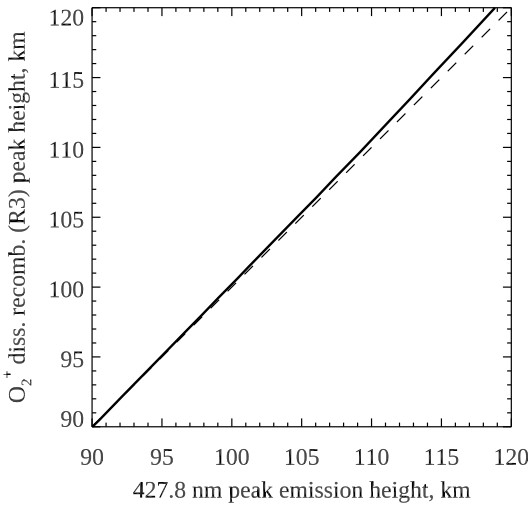

**Figure 12.** Modelled peak height of the production rate of $O(^1S)$ by dissociative recombination of $O_2^+$ (Reaction R3) against the blue 428.8 nm peak emission height (electron impact ionisation of $N_2$). The dashed line marks where the two rates would be equal.

(R5) dominating from 100 km to 116 km (the upper altitude limit of their measurements). Our work and observations support this conclusion, except that in our model the production of $O(^1S)$ by dissociative recombination of $O_2^+$ is too slow compared to energy transfer from $N_2(A)$ at low altitudes. We currently do not have a solution to this discrepancy, but in the following subsections we consider whether the modelled Reaction R3 is too slow, Reaction R5 is too fast, or there is an $O(^1S)$ production
mechanism missing from the Southampton ionospheric model.

### 5.2.1 $O(^1S)$ from dissociative recombination of $O_2^+$

For dissociative recombination of $O_2^+$ the model uses a quantum yield of $O(^1S)$ of 0.05 from Kella et al. (1997), with a temperature dependent rate coefficient from the review by Sheehan and St.-Maurice (2004). The rate coefficient is given by:

$$1.95 \times 10^{-13}(T_e/300)^{-0.70} \text{ m}^3\text{s}^{-1} \tag{1}$$

where $T_e$ is the electron temperature in Kelvin. This rate coefficient was measured by Mehr and Biondi (1969) and agrees exactly with that found by Alge et al. (1983). Peverall et al. (2001) found a rate 23% larger (with the same temperature dependence) although their quantum yield of $O(^1S)$ was lower than that of Kella et al. (1997) (but with an electron energy dependence). There is little evidence in the literature for a substantially larger rate coefficient than this. Abreu et al. (1983) used observations of airglow to determine the quantum yield of $O(^1S)$ and found values between 0.09 and 0.23, with a dependence
on the ratio of electron density to O density (greater yield with larger ratio) from which they inferred a dependence on the vibrational excitation of the $O_2^+$. This vibrational dependence was also found in laboratory measurements by Petrignani et al. (2005), who found quantum yields of 0.06, 0.14 and 0.21 for vibrational levels $v = 0, 1, 2$ respectively. The Southampton model does not track individual vibrational levels of $O_2^+$, but the $O(^1S)$ quantum yield could be increased from 0.05 to reflect the av-





erage value across an assumed vibrational distribution. Using a value of 0.2 in the model does improve the fit to observations
below 110 km, although the modelled height difference is still larger than observations below 100 km.

The production rate of $O(^1S)$ through dissociative recombination (R3) could also be underestimated as a consequence of an underestimated $O_2^+$ density, either because the ionisation rate is too slow or because the $O_2$ mixing ratio is too low. We note that the $N_2$ and O densities were reduced in the latest version of MSIS, but Emmert et al. (2021) conclude that the $O_2$ density is more uncertain due to an absence of suitable data in the middle thermosphere.

### 5.2.2   $O(^1S)$ excited by energy transfer from $N_2(A)$

The production of $O(^1S)$ through energy transfer from $N_2(A)$ depends not only on the rate of the energy transfer itself, but also on the production and loss rates of the long-lived $N_2(A)$. Production of $N_2(A)$ is by electron impact excitation, but cascading from the $B^3\Pi_g$ state via the First Positive transition is the dominant production mechanism, and $B^3\Pi_g$ is itself fed by cascading from the $C^3\Pi_u$ and $W^3\Delta_u$ states. Since these cascade transitions are prompt, to calculate production of $N_2(A)$ the model sums

electron impact excitation to the $A^3\Sigma_u^+$, $B^3\Pi_g$, and $W^3\Delta_u$ states, with half of the excitation to the $C^3\Pi_u$ state also included. For $N_2$ electron impact excitation the model uses the "J05M09" cross sections of Yonker and Bailey (2020), which gave a marginally better agreement with our observations than the cross sections of Itikawa (2006). $N_2(A)$ is lost through quenching by $O_2$ and O with rates from Gattinger et al. (1996) and Piper (1982), respectively, as well as emission in the Vegard-Kaplan (A–X) band system (Einstein coefficient 0.38 s$^{-1}$, from Piper (1993)). Quenching by $O_2$ can indirectly lead to production

of $O(^1S)$ via $O_2^*$, through Reaction R6, although $O_2^*$ is also quenched. During energetic precipitation ($>\sim$10 keV), $O(^1S)$ production by energy transfer from $N_2(A)$ peaks above the $N_2^+$ 1N (427.8 nm) emission since quenching of $N_2(A)$ by $O_2$ at low altitudes raises the peak height of the $N_2(A)$ number density. Below the mesopause the $O(^1S)$ production by energy transfer also falls off with the O number density.

For the $N_2$ energy transfer reaction (R5) we use the rate coefficient ($2.8 \times 10^{-17}$ m$^3$s$^{-1}$molecule$^{-1}$) and $O(^1S)$ quantum

yield (0.75) of Piper (1982). The larger (by 25%) rate coefficient of Thomas and Kaufman (1985) was used by Yonker and Bailey (2020), but using that coefficient in our model would increase the dominance of energy transfer from $N_2(A)$ in the production of $O(^1S)$ at low altitudes and would therefore make the match to observations worse. Hill et al. (2000) proposed a temperature-dependent rate coefficient for Reaction R5, although Yonker and Bailey (2020) argued against a temperature dependence. Replacing the rate from Piper (1982) with that from Hill et al. (2000) led to a marginal improvement in the match between the

model and observations at high electron precipitation energies, but made the match considerably worse at high altitudes for low energy precipitation.



### 5.2.3 Possible pathways for producing $O(^1S)$ missing from the Southampton ionospheric model

Besides overestimated or underestimated rates, there may be sources of $O(^1S)$ completely missing from the Southampton ionospheric model which could explain the discrepancy with our observations at the lowest altitudes. We briefly consider two
possiblities here, represented by:

$$N(^2D) + NO \rightarrow N_2 + O \tag{R7}$$

$$O_2 + e^- \rightarrow O^+ + O + 2e^- \tag{R8}$$

Reaction R7 is thought to occur in the upper atmoshere (e.g. Barth, 1992) and the production of $O(^1S)$ is energetically
allowed, but the quantum yield of $O(^1S)$ is not yet known (Fox and Hać, 2018). The Southampton model uses a rate coefficient for this reaction from Lin and Kaufman (1971) with all O being produced in the ground state. Similarly, electron impact dissociative ionisation of $O_2$ in which one of the products is $O^+$ (R8) is also included in the model (as a distinct reaction to R2), but not as a source of $O(^1S)$. No branching ratio for the production of $O(^1S)$ via R8 is available.

Modelled altitude profiles of the rates of R7 and R8 are plotted in Figure 13 in solid red and dotted blue, respectively, for
a Gaussian electron precipitation spectrum at 25 keV with a flux of 20 mW m$^{-2}$ running for 12 s. R3 (solid cyan) and R5 (dashed magenta) are reproduced from Figure 11 for comparison. The rate of R7 is several orders of magnitude too small to have a significant effect on the auroral peak emission altitude, and peaks at higher altitude than $O_2^+$ dissociative recombination, so we discount that reaction. In contrast, the rate of R8 is approximately twice the rate of $O_2^+$ dissociative recombination (R3) at the peak and therefore may be significant if the quantum yield of $O(^1S)$ is not too low.

Further work is required to evaluate the possible contributions of these, and other, mechanisms to the production of $O(^1S)$ in aurora, ideally including experimental determination of the relevant rate coefficients, cross sections and quantum yields.

## 6   Conclusions

The typical peak emission height of both green 557.7 nm aurora and blue 427.8 nm aurora is about 114 km (113.8 km for green and 113.5 km for blue, according to our large set of observations from Lapland in 2000–2007). When the emissions peak
below this typical height they peak at a very similar height, with the green very slightly (∼1 km) above the blue, on average. When they peak at higher altitudes, the blue usually peaks above the green.

During low energy electron precipitation, the dominant mechanism producing the $O(^1S)$ responsible for the green emission is energy transfer from $N_2(A)$. The green line peaks at lower altitude than the blue line because the production rate of $O(^1S)$ (green line) depends on the product of $N_2$ and O number densities, which both decrease to higher altitude, whereas electron impact
production of $N_2^+$ (blue line) depends on the number density of only $N_2$. The height difference between the two emissions is sensitive to the primary electron spectrum shape as well as the duration of the precipitation, but on average reaches about 10 km when the green line peaks at 145 km.



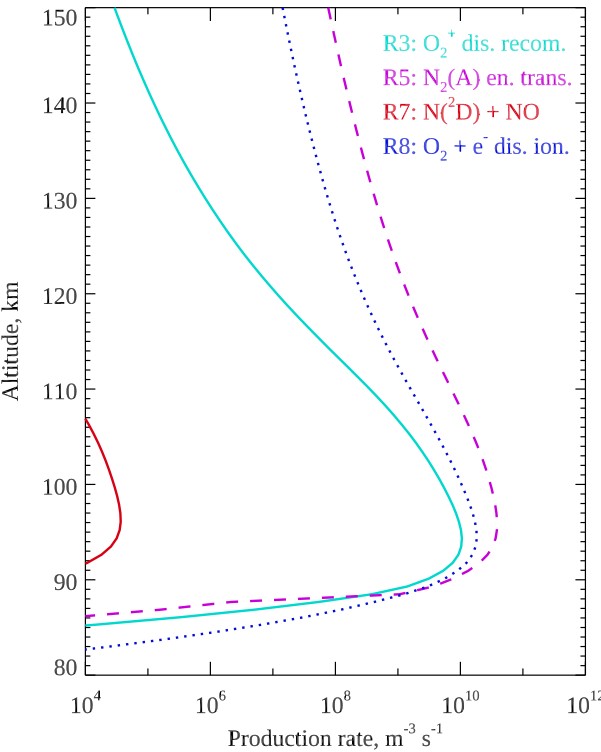

**Figure 13.** Modelled rate profiles after 12 s of Gaussian electron precipitation at 25 keV. The rates of R3, R5, R7 and R8 are plotted in solid cyan, dashed magenta, solid red and dotted blue, respectively. R3 and R5 are production of $O(^1S)$ and are exactly as plotted in Figure 11. R7 and R8 are production of O in any state.

During high energy precipitation, the dominant mechanism producing $O(^1S)$ at low altitudes must not be energy transfer from $N_2(A)$. Dissociative recombination of $O_2^+$ seems the most likely candidate, but in that case the Southampton ionospheric

model underestimates the rate of this process, overestimates the rate of energy transfer from $N_2(A)$ to $O(^1S)$, or a combination of both. We tentatively suggest that electron impact dissociative ionisation of $O_2^+$ in which $O^+$ is one of the fragments may be an additional source of $O(^1S)$ missing from models, although its inclusion may not be enough to bring the Southampton model in line with our observations, and other mechanisms may also be missing.

A comparison between our model results and observations suggests that the auroral electron precipitation energy spectrum

is typically Gaussian when the characteristic energy is above about 4 keV, but closer to Maxwellian at lower characteristic energies.

*Data availability.* MIRACLE ASC quicklook data are available at (https://space.fmi.fi/MIRACLE/ASC/), full resolution image data can be requested from FMI (kirsti.kauristie@fmi.fi).



*Author contributions.* DKW performed the data analysis. All authors contributed to the discussion of the results and the writing of the paper.

*Competing interests.* The authors declare no competing interests.

*Acknowledgements.* This work was financially supported by the Academy of Finland (grant numbers 128553 and 268260). DKW is supported by a Natural Environment Research Council (NERC) Independent Research Fellowship (NE/S015167/1). The work by NP is supported by the Research Council of Norway under CoE contracts 223252 and 287427.

The MIRACLE network is operated as an international collaboration under the leadership of the Finnish Meteorological Institute. The
authors thank M. Syrjäsuo, S. Mäkinen, J. Mattanen, A. Ketola, and C.-F. Enell for careful maintenance of the MIRACLE all-sky cameras and data flow. The $F10.7$ and $Ap$ data were obtained from the National Geophysical Data Center, Boulder. The authors thank L. Juusola for providing IL data.



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
