# Peer review of "The altitude of green OI 557.7 nm and blue $\mathrm{N_2^+}$ 427.8 nm aurora"

_Annales Geophysicae, 2022_

## Author Response (AR1)

**Whiter et al., The altitude of green OI 557.7 nm and blue $N_2^+$ 427.8 nm aurora, submitted to Ann. Geophys.**

**Response to reviewers**

Reviewer 1

We thank the reviewer for the careful and positive evaluation of our manuscript. Here we address the two points raised.

(1) Effects of dynamics

We agree that the motions of O and N2+ will be different, and that N2+ may be transported upwards and then resonantly scatter sunlight in the blue line at 427.8 nm. However, it is important to note that the blue emission is prompt, and therefore in general in aurora is produced at the point of ionisation, and so the peak emission altitude is related to the N2 altitude profile, not the N2+ altitude profile. We considered whether resonant scattering could influence our statistical results by examining the solar zenith angle and magnetic local time distributions at the times when the blue emission was observed at high altitude compared to the green, but found that resonant scattering cannot explain the observations (see Section 3.1 of the manuscript).

We added text at line 119:
The auroral emission is prompt and produced at the point of ionisation, and therefore is related to the altitude profile of the neutral $N_2$ density. Blue line emission originating through resonance scattering is instead related to the $N_2^+$ ion density, so is enhanced by soft electron precipitation producing ground state $N_2^+$ above the shadow height, and may also be influenced by ion transport.

The "bend" at 120 km altitude is reproduced in our model, and seems most likely to be related to the shape of the neutral density profiles, with a change in scale height above the mesopause. Figure 7 shows that below ~120 km the rate of change of peak emission height with precipitation energy is much less than above 120 km, due to the relatively rapid neutral density increase going further down. The balance between the different production mechanisms of O(1S) changes with altitude as the composition changes.

(2) Shape of the modelled spectrum

We used Gaussian and Maxwellian shaped spectra simply because it has been quite standard to use those shapes – Gaussian typically represents discrete aurora while Maxwellian represents more diffuse aurora, see e.g. Strickland et al., 1993, doi:10.1029/93JA01645 and Tanaka et al., 2006, doi:10.1029/2006JA011744. We added these references and text at line 141:
In this work we use Gaussian or Maxwellian shaped electron precipitation spectra, as has been standard in auroral physics (e.g. Strickland et al., 1993; Tanaka et al., 2006).

Reviewer 2

We thank the reviewer for the careful and positive evaluation of our manuscript.